# The impact of physical exercise on adolescents' social-emotional competence: The chain mediating role of social support and peer relationships

Qin Zeng[1], Pengfei Wen[1]*, Kelei Guo[2]*

**1** Guangzhou Sport University, Guangzhou, Guangdong, China, **2** Zhaoqing University, Zhaoqing, Guangdong, China

* 275695156@qq.com (PFW); guokelei20040328@163.com (KLG)

## Abstract

This study aims to explore the relationship between physical exercise and adolescents' social-emotional competence, and analyze the chain mediating effects of social support and peer relationships. Data from 316 adolescents were collected through questionnaires, and measurements were conducted on four aspects: physical exercise, social support, peer relationships, and adolescents' social-emotional competence. The results showed that physical exercise had a significant positive predictive effect on adolescents' social-emotional competence, with social support and peer relationships playing significant mediating roles between physical exercise and adolescents' social-emotional competence. Three specific mediating pathways were identified: (1) the independent mediating effect of social support between physical exercise and adolescents' social-emotional competence; (2) the independent mediating effect of peer relationships between physical exercise and adolescents' social-emotional competence; and (3) the chain mediating effect of social support and peer relationships between physical exercise and adolescents' social-emotional competence. The findings reveal the intrinsic mechanisms through which physical exercise influences adolescents' social-emotional competence, while also providing theoretical support and practical evidence for promoting the development of adolescents' social-emotional competence.

## 1. Introduction

With the deepening of contemporary social transformation, the cultivation of non-cognitive skills has become a critical dimension of adolescents' holistic development [1]. Social-emotional competence serving as a core assessment indicator of non-cognitive skills [2], attracts continuing research attention from the international academic community. In its policy framework Education 2030: The Future of Education

**Data availability statement:** All relevant data are within the paper and its Supporting Information files.

**Funding:** This work was supported by The Educational Science Planning of Guangdong Province project of China (Grant number 2023GXJK354 to QZ), Higher Education Teaching Research and Reform of Guangdong Province project of China and Guangdong Province Philosophy (to QZ) and Social Science Planning Project (Grant number GD23YDXZTY02 to QZ).

**Competing interests:** The authors have declared that no competing interests exist.

and Skills, the Organization for Economic Co-operation and Development (OECD) systematically established a three-dimensional competency model encompassing cognitive, social-emotional, and practical skills, explicitly defining them as essential core competencies for 21st-century adolescents [3]. The global influence of this framework, along with its cross-national SSES (Study on Social and Emotional Skills) research project conducted across 38 member countries, further underscores the prominent role of social-emotional competence cultivation in international educational strategies.

Social-emotional competence refer to the comprehensive ability to acquire and appropriately apply knowledge, skills, and attitudes to regulate and control emotions, make effective judgments in decision-making, communicate constructively with others, and resolve problems in complex environments [4]. There are five dimensions: self-awareness, self-management, social awareness, interpersonal skills, and responsible decision-making [5]. Studies indicate that strong social-emotional competence help adolescents improve academic performance [6], enhance well-being [7], promote mental health [8], and reduce problematic behaviors [9]. However, adolescents with lack of social-emotional competence are prone to emotional regulation difficulties and social disorders [10], which can result in mental health problems such as despair, anxiety, and low self-esteem [11]. Globally, the development of adolescents' social-emotional competence remains suboptimal. The world health organization's 2024 report on adolescent mental health revealed that 14% of adolescents worldwide suffer from varying degrees of mental disorders (e.g., anxiety, depression, social difficulties), while 8.3% exhibit behavioral disorders (e.g., impulsivity, aggression) [12]. Many adolescents lack empathy, decision-making skills, and conflict-resolution abilities, with severe issues like school bullying, substance abuse, and suicide remaining prevalent worldwide—all closely linked to deficiencies in social-emotional competence [13].

Numerous studies indicate that physical exercise may have a positive impact on social-emotional competence. Adolescents' participation in physical exercise can effectively develop their social interaction skills [14], enhance cooperation, mutual assistance, and team awareness [15], improve emotional regulation abilities [16], and influence the development of their self-awareness by enhancing physical literacy [17]. However, most of these studies focus on the impact mechanisms of physical exercise on single dimensions of social-emotional competence, such as emotional regulation and interpersonal communication, lacking research on the overall structural correlation of adolescents' social-emotional competence. One study showed that physical exercise can help adolescents obtain social support from important figures such as friends and family, and promote the positive development of peer relationships [18]. Furthermore, the development of peer relationships was proven to be positively correlated with social-emotional competence, and this relationship is more pronounced among adolescent groups [19]. This raises the question: Can physical exercise affect adolescents' social-emotional competence through the chain mediating effect of social support and peer relationships? Therefore, this study intends to construct a multidimensional analysis framework of "physical exercise - mediating variables

- social-emotional competence". By systematically analyzing the chain mediating effect of social support and peer relationships on social-emotional competence, we aim to explore the influencing factors and operating mechanisms of adolescents' social-emotional competence from multiple perspectives, thereby promoting the healthy personality development of contemporary adolescents.

## 1.1. The relationship between physical exercise and adolescents' social-emotional competence

As one of the most significant leisure activities in modern life, physical exercise effectively enhances individuals' physical fitness, promotes mental and physical health, and sustains various bodily capabilities [20]. Research indicates that adolescents who receive positive feedback—such as achieving excellent performance, gaining recognition from others, or acquiring new skills—experience a significant boost in self-confidence and self-worth, thereby fostering the development of self-awareness [21]. Empirical data demonstrate that adolescents who engage in long-term moderate-to-vigorous physical exercise score higher in the cognitive empathy dimension [22], suggesting that physical exercise may enhance social awareness through strengthening empathic abilities [23]. The improvement of emotional regulation skills through physical exercise helps alleviate negative emotions like anxiety and depression, laying the foundation for advancing self-management abilities [24]. Furthermore, frequent team collaboration and social interactions in sports settings not only heighten individuals' sensitivity to social norms but also refine interpersonal communication skills, optimizing relationship management [25]. Additionally, the activation of higher-order cognitive functions (e.g., strategic analysis, conflict resolution) through physical exercise enables adolescents to more rationally evaluate behavioral consequences in complex scenarios, ultimately fostering responsible decision-making [26].

In summary, this study posits that physical exercise may exert specific mechanisms of influence on adolescents' social-emotional competence, leading to Hypothesis 1: Physical exercise positively promotes social-emotional competence.

## 1.2. The mediating role of social support between physical exercise and social-emotional competence

One of the mediating mechanisms explored in this study is the mediating effect of social support. Social support typically refers to the emotional and material assistance individuals receive from their social relationships [27], encompassing two dimensions: perceived social support (subjective judgments and evaluations of reliable relationships) and actual social support (tangible assistance provided by others) [28]. Research indicates that the ability to perceive social support is strongly influenced by personal factors, including well-being and psychological control source. Through scientific physical exercise, individuals can enhance well-being [29] and develop a greater inclination toward an internal locus of control when responding to event [30], thereby increasing their likelihood of recognizing supportive behaviors from others in daily life [31]. Additionally, physical exercise serves as a vital pathway to elevate adolescents' social support levels [32]. Regular participation in physical exercise is associated with heightened capacities to both provide and receive material, emotional, and spiritual support. It also expands social networks and enhances social cognition, enabling individuals to access more social support [33]. Benedetti (2011) further found that individuals with a history of physical exercise are more likely to engage in social or altruistic activities, which not only enrich their social connections but also amplify their social support [34].

Moreover a close relationship exists between social support and social-emotional competence [35]. The social support acquired by adolescents promotes positive development across sub-dimensions of social-emotional competence, thereby driving the overall enhancement of their social-emotional competence [36]. The buffering effect model theory posits that social support mitigates adverse emotions such as anxiety, fear, and depression caused by stress, strengthens emotional regulation abilities, improves mental health [37], and significantly predicts interpersonal communication skills [38]. Particularly during adolescence, when individuals develop a heightened sense of autonomy and self-esteem, external support becomes critical to the formation of self-awareness [39]. High levels of social support enable adolescents to affirm their self-worth, elevate self-perception [40], and consequently exhibit stronger social-emotional competence.

Based on this, Hypothesis 2 is proposed: Social support plays a mediating role between physical exercise and social-emotional competence.

## 1.3. The mediating role of peer relationships between physical exercise and social-emotional competence

Another mediating mechanism explored in this study is the mediating effect of peer relationships. Peer relationships refer to interpersonal connections established through communication among peers or individuals at similar psychological developmental stages [41]. Research indicates that regular participation in physical exercise helps adolescents build healthy peer relationships and improve the quality of friendships [42]. During this process, mutual encouragement, shared experiences of overcoming challenges, and collective achievements foster a positive peer atmosphere [43], which plays an irreplaceable role in maintaining healthy peer relationships [44]. Additionally, physical exercise provides adolescents with a platform for peer interaction and communication. In team sports, for instance, adolescents often need to collaborate, communicate, and support one another, facilitating the positive development of peer relationships [45].

Moreover, maintaining healthy peer relationships requires mutual respect and cooperation, offering individuals opportunities to interact with peers who share similar thoughts, feelings, motivations, and intentions. This enhances interpersonal sensitivity, allows adolescents to experience intimacy and mutual understanding, and cultivates essential social skills, thereby improving social awareness and empathy [46]. In the development of peer relationship, adolescents need to correctly deal with the relationship between individuals and others, and between individuals and groups, so as to realize the construction of self-cognition structure and promote the development of individual self-awareness to a certain extent [47]. Yang (2017) found that adolescents with strong peer relationships excel in emotional regulation, social sensitivity, understanding others, altruistic behaviors, sharing, caring, and providing support [48]. Furthermore, studies demonstrate a strong correlation between healthy peer relationships and high levels of social competence [49], highlighting their critical role in the development of social-emotional competence below [50].

Based on this, Hypothesis 3 is proposed: Peer relationships play a mediating role between physical exercise and social-emotional competence.

## 1.4. The chain mediating effect of social support and peer relationships

Studies reveal a significant positive correlation between social support and peer relationships [51]. Adolescents who perceive recognition, understanding, and respect in social interactions are more likely to develop positive peer relationships [52]. This benign interaction stems from individuals' ability to perceive social support. Adolescents with strong perception skills not only identify positive environmental cues but also proactively build harmonious relationships, reducing conflicts and fostering sustainable positive interactions [53]. Support from family, school, and community further promotes adolescents' communication with external environments, enhancing peer relationships [54]. Higher levels of social support also facilitate the formation of interactive and cooperative peer relationships [55]. Guyer (2015) provided physiological evidence that individuals receiving greater familial support tend to be more popular among peers. Enhanced integration of the prefrontal-limbic system enables adolescents to flexibly navigate social challenges, reduce sensitivity to peer rejection, and strengthen peer relationships [56].

Based on this, Hypothesis 4 is proposed: Social support and peer relationships exhibit a chain mediating effect between physical exercise and social-emotional competence.

Summary: To investigate the intrinsic mechanisms linking physical exercise and social-emotional competence, this study constructs a chain mediation model (Fig 1) and aims to validate the following: (1) physical exercise positively predicts adolescents' social-emotional competence; (2) social support and peer relationships independently mediate the relationship between physical exercise and social-emotional competence; (3) social support and peer relationships exhibit a chain mediating effect between physical exercise and social-emotional competence.(Fig 1. Conceptual Framework Diagram).

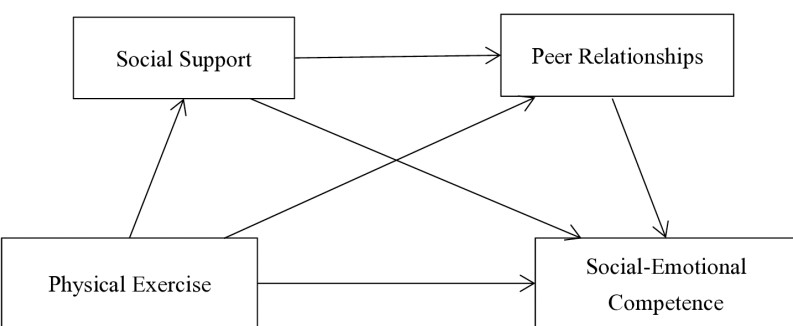

**Fig 1. Conceptual Frmework Diagrom.**

## 2. Research methods

### 2.1. Participants

This study employed a convenience sampling method to select participants aged 10–19 (WHO defines adolescence as 10–19 years [57]) from three middle schools in Shenzhen and Guangzhou, Guangdong Province, China. Adolescents were surveyed between November 1 and November 30, 2024. The inclusion criteria for subjects were as follows: (1) Students aged 10–19; (2) Physically healthy with no motor impairments; (3) No cognitive impairments and able to accurately understand questionnaires and instructions. Individuals not meeting these criteria were excluded. A total of 427 integrated questionnaires containing four scales were distributed, yielding 427 responses. After excluding 111 invalid responses (e.g., incomplete or duplicate entries), 316 valid datasets were retained, representing a 74% validity rate. The participants' mean age was 14.73±1.40 years, including 164 boys (51.90%) and 152 girls (48.10%), with 108 senior high school students (34.18%) and 208 junior high school students (65.82%). Paper questionnaires were administered by the research team during self-study sessions in classroom settings. Before completing the questionnaires, researchers thoroughly explained the purpose, provided clear instructions, and addressed participants' questions or concerns. Subjects were required to complete an integrated questionnaire comprising four scales: The Physical Activity Rating Scale, The Adolescent Social Support Scale, The Student Peer Relationships Scale, and The Social-Emotional Competence Questionnaire, totaling 66 items with an estimated completion time of 10 minutes.

Before conducting the questionnaire survey, to standardize data collection, all researchers received centralized training in advance. During fieldwork, they monitored participants' completion process, informed subjects of relevant details, obtained their informed consent, and secured their written consent. As participants were minors, written consent was also obtained from their guardians. Additionally, the study design was approved by the Human Research Ethics Committee of Guangzhou Sport University (Approval No.: 2024LCLL-71). Questionnaire instructions emphasized anonymity, assuring respondents that there were no 'right' or 'wrong' answers, clarifying that data would be used solely for scientific research, and specifying the estimated completion time.

### 2.2. Variable measurement

**2.2.1. Measurement of physical exercise.** Physical exercise was assessed using the Physical Activity Rating Scale developed by Hashimoto(1990) [58] and revised by Liang, D. Q(1994) [59]. This scale quantifies exercise levels through three dimensions: exercise intensity, exercise duration, and exercise frequency. The total exercise level was calculated as Physical Exercise Level = Duration × Intensity × Frequency. Physical Exercise Level = Duration × Intensity × Frequency. The scale comprises 3 items rated on a 5-point Likert scale. Intensity and frequency were scored 1–5, while duration was scored 0–4. Higher total scores indicate greater exercise volume. Previous studies have confirmed that the scale is applicable to adolescent populations [60,61]. The Cronbach's alpha coefficient for this scale was 0.732.

**2.2.2. Measurement of social support.** Social support was measured using the Adolescent Social Support Scale developed by Ye, Y. M(2008) [62]. This scale evaluates three dimensions: objective support, subjective support, and support utilization. The 17-item scale uses a 5-point Likert scale (1 = "strongly disagree " to 5 = "strongly agree"). Total scores reflect overall social support, with higher scores indicating greater support. Previous studies have confirmed that the scale is applicable to adolescent populations [63,64]. The Cronbach's alpha coefficient was 0.950.

**2.2.3. Measurement of peer relationships.** Peer relationships were assessed using the Student Peer Relationship Scale developed by Asher, S. R (1985) [65]and revised by Zhang, Y. L(2008) [66]. It includes three dimensions: acceptance, rejection, and loneliness. The 16-item scale employs a 4-point Likert scale (1 = "strongly disagree" to 4 = "strongly agree"). Items 2, 4, 7, 8, 9, 11, 12, 13, 14, and 16 were reverse-scored. Higher total scores indicate better peer relationships. Previous studies have confirmed that the scale is applicable to adolescent populations [67,68]. The Cronbach's alpha coefficient was 0.865.

**2.2.4. Measurement of social-emotional competence.** Social-emotional competence was evaluated using the Social-Emotional Competence Questionnaire jointly developed by the Collaborative for Academic, Social, and Emotional Learning (CASEL) and the UNICEF-Ministry of Education "Social and Emotional Learning (SEL)" project, revised by Guo, Y (2019) [69]. The scale covers five dimensions: self-awareness, self-management, social awareness, interpersonal skills, and responsible decision-making. The 30-item scale uses a 5-point Likert scale (1 = "strongly disagree" to 5 = "strongly agree"). Higher total scores reflect stronger social-emotional competence. Previous studies have confirmed that the scale is applicable to adolescent populations [70,71]. The Cronbach's alpha coefficient was 0.956.

## 2.3. Data processing

All statistical analyses were conducted using SPSS 26.0, Amos 24.0, and the SPSS macro-PROCESS plugin developed by Hayes (2013). First, common method bias (CMB) was tested with SPSS 26.0. Pearson correlation analysis was then performed to examine relationships among physical exercise, social support, peer relationships, and social-emotional competence. Finally, the PROCESS macro plugin was employed to test the mediating effects of social support and peer relationships between physical exercise and social-emotional competence, as well as the chain mediating effect of social support and peer relationships.

## 3. Results and analysis

### 3.1. Common method bias test

As self-reported data collection may introduce common method bias (CMB), necessary controls were implemented during the survey, such as emphasizing anonymity, clarifying that data would be used solely for research, and incorporating reverse-scored items. Harman's single-factor test was applied to the collected data. Unrotated exploratory factor analysis (EFA) extracted 11 factors with eigenvalues >1, with the largest factor explaining 34.72% of the variance (<40%). Confirmatory factor analysis (CFA) was conducted by extracting a common factor from all variables and loading all items onto it. The model showed poor fit indices: $\chi^2/df = 3.84$, CFI = 0.56, TLI = 0.55, IFI = 0.57, RMSEA = 0.10, and SRMR = 0.10, indicating no single factor explained most variance. Thus, no severe CMB was present.

### 3.2. Descriptive statistics and correlation analysis

As shown in Table 1, physical exercise, social support, peer relationships, and social-emotional competence all demonstrate statistically significant positive correlations. Among these, social support shows the strongest correlation with social-emotional competence (r = 0.72). Peer relationships exhibit the second-highest correlation with social-emotional competence (r = 0.51), while their correlation with social support is relatively weaker (r = 0.42). Physical exercise displays moderately positive correlations with social-emotional competence (r = 0.43), peer relationships (r = 0.38), and social

**Table 1. Means, Standard Deviations, and Correlation Coefficients of Variables.**

| Variable | M | SD | 1 | 2 | 3 | 4 |
|---|---|---|---|---|---|---|
| 1.Physical Exercise | 2.59 | 0.94 | 1 | | | |
| 2.Social-Emotional Competence | 3.65 | 0.66 | 0.43** | 1 | | |
| 3.Social Support | 3.66 | 0.82 | 0.30** | 0.72** | 1 | |
| 4.Peer Relationships | 2.79 | 0.47 | 0.38** | 0.51** | 0.42** | 1 |

N = 316; * p<0.05, ** p<0.01.

support (r = 0.30). These data provide preliminary evidence suggesting that social support and peer relationships may play mediating roles in the relationship between physical exercise and adolescents' social-emotional competence.

### 3.3. Mediating effects of social support and peer relationships

The SPSS macro-PROCESS (Hayes, 2013) was used to conduct bootstrap-based mediation analysis, a total of 5,000 Bootstrap samples were used to test the mediating effects, generating bias-corrected (BC) 95% confidence intervals, specifically testing the chain mediation model (Model 6). As shown in Table 2, physical exercise significantly and positively predicted adolescents' social-emotional competence ($\beta = 0.14$, $p < 0.01$), supporting Hypothesis 1. After incorporating social support and peer relationships into the regression model, physical exercise significantly and positively predicted social support ($\beta = 0.16$, $p < 0.01$) and peer relationships ($\beta = 0.08$, $p < 0.01$); social support significantly and positively predicted peer relationships ($\beta = 0.18$, $p < 0.01$); and social support ($\beta = 0.83$, $p < 0.01$) and peer relationships ($\beta = 0.52$, $p < 0.01$) significantly and positively predicted social-emotional competence. Even after accounting for these mediators, physical exercise retained a significant direct positive effect on social-emotional competence ($\beta = 0.34$ Among, $p < 0.01$).

The results of mediating effect magnitude analysis (Table 3 and Fig 2) indicate that social support and peer relationships significantly mediate the relationship between physical exercise and adolescents' social-emotional competence, with a total standardized indirect effect of 0.194 This mediation effect is specifically composed of three indirect pathways: Indirect Effect 1 (Effect = 0.14): Physical Exercise → Social Support → Social-Emotional Competence; Indirect Effect 2 (Effect = 0.04): Physical Exercise → Peer Relationships → Social-Emotional Competence; Indirect Effect 3 (Effect = 0.02): Physical Exercise → Social Support → Peer Relationships → Social-Emotional Competence. These three indirect effects accounted for 40.60%, 12.84%, and 4.48% of the total effect, respectively. All 95% bootstrap confidence intervals for these indirect effects excluded zero, confirming their statistical significance. Thus, Hypotheses 2, 3, and 4 are fully supported. (Fig 2. The Chain Mediating Model of Physical Exercise and Social-Emotional Competence).

**Table 2. Analysis of Regression Relationships Variables.**

| Effect | Item | Effect | SE | t | p | LLCI | ULCI |
|---|---|---|---|---|---|---|---|
| Direct Effect | Physical Exercise →Social-Emotional Competence | 0.14 | 0.03 | 4.55 | 0.000 | 0.08 | 0.20 |
| | Physical Exercise →Social Support | 0.16 | 0.03 | 5.47 | 0.000 | 0.11 | 0.22 |
| | Physical Exercise →Peer Relationships | 0.08 | 0.02 | 5.44 | 0.000 | 0.05 | 0.11 |
| Indirect Effect Process | Social Support →Peer Relationships | 0.18 | 0.03 | 6.45 | 0.000 | 0.12 | 0.23 |
| | Social Support →Social-Emotional Competence | 0.83 | 0.06 | 14.67 | 0.000 | 0.72 | 0.95 |
| | Peer Relationships →Social-Emotional Competence | 0.52 | 0.11 | 4.68 | 0.000 | 0.30 | 0.73 |
| Total Effect | Physical Exercise →Social-Emotional Competence | 0.34 | 0.04 | 8.33 | 0.000 | 0.26 | 0.41 |

**Table 3. Mediation Effect Analysis of Physical Exercise and Social-Emotional Competence.**

| Item | Effect | Boot SE | LLCI | ULCI | Relative Mediation Effect |
|---|---|---|---|---|---|
| Physical Exercise →Social Support →Social-Emotional Competence | 0.14 | 0.03 | 0.08 | 0.20 | 40.60% |
| Physical Exercise →Peer Relationships →Social-Emotional Competence | 0.04 | 0.02 | 0.02 | 0.08 | 12.84% |
| Physical Exercise →Social Support→ Peer Relationships →Social-Emotional Competence | 0.02 | 0.01 | 0.01 | 0.03 | 4.48% |
| Total Mediation Effect | 0.19 | 0.03 | 0.13 | 0.26 | 57.91% |

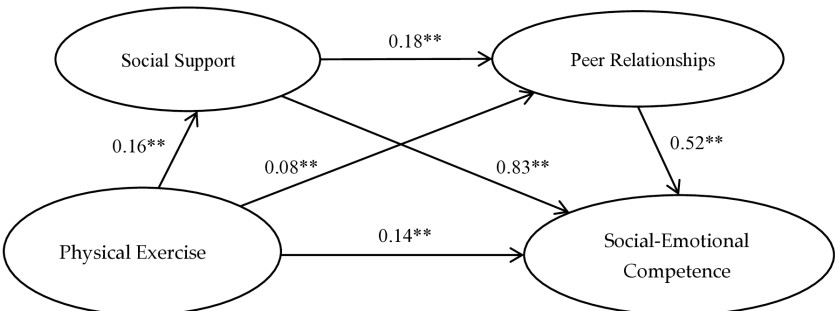

**Fig 2. The Chain Mediating Model of Physical Exercise and Social-Emotional Competence.**

## 4. Discussion

### 4.1. Physical exercise and social-emotional competence

This study found that physical exercise significantly and positively predicts social-emotional competence, validating Hypothesis 1. From the perspective of the Five-Factor Personality Model, social-emotional competence encompasses five dimensions: task competence, emotional regulation, collaborative skills, interpersonal abilities, and openness [72]. First, the process of overcoming challenges during physical exercise effectively cultivates stress resilience and problem-solving skills while significantly enhancing self-control [73] and perseverance [74], thereby advancing task competence. Second, physical exercise stimulates the secretion of hormones such as endorphins, fostering positive emotions and serving as an outlet for stress relief, which improves emotional regulation [59]. Third, the inherently social nature of physical exercise [75] necessitates communication and interaction, particularly in team sports. This process strengthens collaborative and interpersonal skills [76]. Finally, physical exercise provides adolescents with a challenging yet cooperative environment where learning new movements, strategies, and peer interactions expands cognitive boundaries, nurtures innovative thinking, and refines problem-solving abilities [77]. Additionally, continuous exposure to new knowledge and skills during physical exercise promotes the development of open-mindedness. Enhanced open-mindedness enables adolescents to flexibly apply learned knowledge in complex situations, demonstrating greater adaptability and creativity [78].

Therefore, physical exercise can significantly and positively predict adolescents' social-emotional competence.

### 4.2. The independent mediating role of social support

Research has confirmed that social support plays a positively mediating role in the impact of physical exercise on social-emotional competence, validating Hypothesis 2, which aligns with previous findings [79]. Adolescents from families that maintain long-term physical exercise habits typically have closer relationships with their parents and are more likely to receive greater familial support [80]. This occurs because adolescents' cognitive abilities are not yet fully developed,

 

and their lower economic status within families makes parental involvement in family physical exercise more focused on conscious and deliberate assistance. Consequently, adolescents receive multidimensional family support, including instrumental support (e.g., sports equipment purchases, fitness investments), informational support (e.g., verbal instruction and behavioral modeling of physical activities), and appraisal support (e.g., encouragement and praise) [81,82]. Furthermore, during physical exercise participation—particularly in team sports or competitive settings—adolescents strengthen the frequency and depth of peer interactions, building trust and belongingness with others [83]. This establishes a foundation for forming long-term interpersonal networks and provides sustainable social support [84], which in turn promotes adolescents' engagement in physical exercise, creating a virtuous cycle [85]. Additionally, social support equips adolescents with essential social skills and emotional assistance while enhancing emotional regulation, thereby indicating healthier psychological states [86]. By earning recognition and support from family, school, and society through physical exercise, adolescents strengthen self-identity, refine social skills, this fosters prosocial behavior in adolescents [87] and enhances their empathic ability [88], and mitigate negative emotions, all of which significantly contribute to advancing their social-emotional competence.

Thus, physical exercise not only directly enhances adolescents' social-emotional competence but also achieves this indirectly by elevating social support.

## 4.3. The independent mediating role of peer relationships

This study also confirms that peer relationships play a positively mediating role in the impact of physical exercise on social-emotional competence, validating Hypothesis 3. Peer relationships can manifest as peer acceptance or peer rejection. Peer acceptance refers to an individual's popularity within a group, and adolescents' popularity in peer groups is influenced by multiple factors, with participation in physical exercise identified as a critical contributor [89]. Weiss (1992) proposed that physical exercise serves as a key pathway for adolescents to gain peer acceptance, highlighting a significant association between physical competence and peer relationships —a phenomenon particularly pronounced among male adolescents. Further research elaborates that active engagement in physical exercise enhances adolescents' popularity within peer groups, effectively fostering the positive development of peer relationships [19]. Moreover, participation in physical exercise enables adolescents to perceive supportive friendships, promotes peer relationships and interpersonal interactions, reduces negative emotions, and improves mental health [90]. Compared to childhood, adolescents spend significantly more time interacting with peers, and physical exercise, as a vital mode of peer interaction, strengthens both dyadic and group-level relationships among adolescents [91]. Furthermore, peer relationships can significantly influence adolescents' social-emotional competence. Sullivan's interpersonal relations theory regards peer relationships as a key factor in individuals' socialization development and healthy personality formation, playing a crucial role in the development of adolescents' self-awareness and interpersonal skills [92]. Individuals with positive peer relationships demonstrate particularly high levels of performance in dimensions such as emotional comprehension, cognitive capacity for emotional expression, and adoption of others' emotional perspectives. More importantly, peer relationships exert a strong positive effect on emotional regulation, enabling individuals to become more adept at adaptive emotional regulation and effectively inhibit impulsive behaviors, thereby promoting the development of social-emotional competence [93].

Therefore, adolescents can cultivate healthy peer relationships through physical exercise. In a supportive peer environment, mutual learning and encouragement create an optimal setting for nurturing social-emotional competence.

## 4.4. Chain mediation model analysis

This study explores the interconnections among physical exercise, adolescents' social-emotional competence, social support, and peer relationships, revealing that physical exercise not only influences social-emotional competence through the single mediators of social support or peer relationships but also promotes its positive development via a chain mediation

pathway (physical exercise→social support→peer relationships→social-emotional competence), thereby validating Hypothesis 4. The findings indicate that social support consistently enhances peer relationships, meaning higher levels of social support facilitate the establishment of healthier peer relationships—a conclusion consistent with prior research on the role of social support in improving peer relationships [55]. As a critical pathway to elevate adolescents' social support [94], regular physical exercise enables individuals to discover shared interests with peers, strengthen interpersonal bonds, and gain recognition and assistance [95]. Higher levels of social support can further promote the development of peer relationships. For instance, familial support enhances functional integration of the prefrontal-limbic system, reducing sensitivity to peer rejection [56]. Teacher support reduces peer conflicts, as teachers provide timely and consistent responses to conflicts, spend time listening to both parties, and help adolescents resolve interpersonal conflicts [96]. Furthermore, peer relationships exhibit a significant positive correlation with self-awareness. Healthy peer relationships aid individuals in developing positive self-evaluation, self-esteem, self-identity, and perceptions of others and the world [97]. The emotional-sharing mechanisms formed among peers also regulate emotional states and alleviate multiple stressors [98], playing a pivotal role in advancing social-emotional competence.

Thus, social support and peer relationships jointly exert a chain mediating effect on the relationship between physical exercise and adolescents' social-emotional competence.

### 4.5. Limitations

This study has several limitations. First, the research design is cross-sectional, which may limit the effectiveness of causal inference. Future studies should combine longitudinal and cross-sectional approaches to better explore intrinsic causal relationships. Second, this study only examined the mediating effects of social support and peer relationships in the relationship between physical exercise and adolescents' social-emotional competence. However, whether other potential variables exist requires verification through subsequent research. Third, the use of self-report methods in this study introduces potential subjective bias. Future research could adopt a combination of external evaluations and self-reports to enhance the objectivity and reliability of data. Fourth, this study sampled adolescents from Guangdong Province, China. Due to influences from their educational and social contexts, the findings' applicability across diverse backgrounds may be limited. Subsequent research should be conducted in multiple countries and varied settings.

### 5. Conclusions

This study constructed a chain mediation model through a cross-sectional survey of adolescents in Guangdong Province, China, to investigate the relationship between physical exercise and adolescents' social-emotional competence. It provides theoretical support and practical evidence for cultivating adolescents' social-emotional competence, yielding the following conclusions:(1) Physical exercise can significantly and positively predict adolescents' social-emotional competence;(2) Social support and peer relationships play a chain mediating role between physical exercise and adolescents' social-emotional competence.

### Supporting information

**S1 Data. This file contains all raw data points collected during the study, including measurements of Physical Exercise, Social-Emotional Competence, and Social Support, Peer Relationships which were used to generate Figures 2 and Table 1,2,3.**
(XLSX)

### Acknowledgments

This thesis was independently written and researched by the author. I would like to express my gratitude to the team members for their assistance in completing this study, and extend my thanks to the adolescents who participated in the survey.

## Author contributions

**Supervision:** Kelei Guo.

**Writing – original draft:** Qin Zeng.

**Writing – review & editing:** Pengfei Wen.

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
