## [Decision Letter · Decision Letter 0]

19 Jun 2025

Dear Dr. zeng,

Thank you for submitting your manuscript to PLOS ONE. After careful consideration, we feel that it has merit but does not fully meet PLOS ONE’s publication criteria as it currently stands. Therefore, we invite you to submit a revised version of the manuscript that addresses the points raised during the review process.

We look forward to receiving your revised manuscript.

Kind regards,

Elena del Pilar Jiménez-Pérez, Ph.D.

Academic Editor

PLOS ONE

Journal Requirements:

 [This work was supported by the Guangdong Provincial Education Science Planning Project (2023GXJK354), the Guangdong Provincial University Ideological and Political Education Research Project (2021GXSZ061) and the 2024 Guangdong Provincial Higher Education Teaching Research and Reform Project, and the 2021 Guangdong Provincial Quality Engineering (Teaching and Research Office of the School Physical Education Curriculum Group); 2021 Guangdong Province Quality Engineering (Provincial First-class Undergraduate Course "School Sports"); 2023 Guangdong Provincial Quality Engineering (Excellent Sports Talent Training Program of Zhaoqing University); Funded by Guangdong Philosophy and Social Science Planning Project (GD23YDXZTY02).]. 

4. Please include a caption for figure 1 and 2.

Reviewers' comments:

Reviewer's Responses to Questions

**Comments to the Author**

1. Is the manuscript technically sound, and do the data support the conclusions?

Reviewer #1: Partly

Reviewer #2: Partly

Reviewer #3: Yes

2. Has the statistical analysis been performed appropriately and rigorously?

Reviewer #1: I Don't Know

Reviewer #2: No

Reviewer #3: Yes

3. Have the authors made all data underlying the findings in their manuscript fully available?

Reviewer #1: No

Reviewer #2: No

Reviewer #3: No

4. Is the manuscript presented in an intelligible fashion and written in standard English?

Reviewer #1: No

Reviewer #2: Yes

Reviewer #3: Yes

Reviewer #1: This study aims to explore the relationship between physical exercise and adolescent socio-emotional competence and to analyze the mediating effects of social support and peer relationships in the chain. An interesting idea, but not a new one. This manuscript could bring an additional novelty, if it presented the details of the research application correctly and completely.

In the Introduction section, I recommend these studies, even though they are applied to athletes, most of whom are adolescents and provide support for the authors' argument:

https://doi.org/10.1177/00315125211005235,

https://doi.org/10.3390/children9050753

https://doi.org/10.7717/peerj.12803,

https://doi.org/10.1177/00315125221135669

https://doi.org/10.3390/ijerph19031696

L151: "A total of 427 questionnaires were distributed ...." ??? Were 427 questionnaires or a single questionnaire applied to 427 subjects? I don't think it would be possible to apply 427 questionnaires!!! I think it's a mistake in the authors' expression. I recommend correct adaptation everywhere in this section and throughout the manuscript. Example here too ” 316 valid questionnaires”???

L154 ”The sample included 164 males and 152 females...” – I think this formulation should be adapted (example, maybe: teenagers and teenage girls), because at the age of 14 subjects cannot be classified as men or women

L155: ” ± 1.405 years.” ?? I think expressing with only 2 decimal places would be ok! That is ” ± 1.40 years.”

I can't find the criteria for inclusion and exclusion of subjects in the study? I recommend highlighting them in the manuscript

L163: ” Physical Activity Rating Scale developed by Hashimoto(1990)” – why such an old instrument? 35 years old? The authors should argue, because today's adolescents have a completely different vision of social aspects, compared to respondents 35 years ago, when was the instrument validated? Was it validated on a specific population? (another question addressed to the authors)

In the "2.Research Methods" section, data should be provided regarding the duration of completion, where were these assessment instruments applied, as a location? How long did it take to complete them? Were the subjects assisted by an adult? Did the subjects understand, accurately, at their age, all the items that the 4 instruments assess? Did they have any questions and to whom did they address them? How were the 4 instruments applied: in a single form or in separate forms? Were the subjects aware that they were completing 4 assessments or did they know that they were completing a single instrument with 4 aspects? Is this section unclear and does not provide criteria for the reproducibility of the study?

The supplementary documents are not in English, which does not allow the reviewers to analyze these aspects (for example, I would like to find out details about Research protocols and Data.xlsx to verify the validity of the data, but I do not know the language used by the authors). I believe that translating the supplementary documents into English is essential, especially since some reviewers do not know the language used by the authors. I do not see the point of sending the main manuscript in English, the platform asks me if the English used by the authors is ok, but I, as a reviewer, cannot analyze most of the documents!!!!!

This study does not allow for data validation or reproducibility... due to the lack of certain details and supporting documents in a language that reviewers are not required to know in this journal.

Reviewer #2: Dear authors,

I have studied an interesting article, with a well-proposed purpose, but with some limitations that require clarification.

Below, I offer my observations and recommendations:

1. The Introduction section should include more aspects of social support through sports activities

2. I ask the authors to argue the relevance of the instruments used, compared to other instruments for evaluating the aspects targeted, which are newer, more recent (The first evaluation instrument is from 1990!!);

3. I recommend that the authors add to the manuscript the criteria for including and excluding subjects in the research group

4. I ask the authors to provide more details in the Research Methods section about how, when, where, how long the practical intervention lasted? (application of the questionnaire or the 4 evaluation instruments). To specify whether the adolescent subjects knew that they would be part of a research that targets several aspects!? The level of education of the adolescents is relevant and should be specified (were all respondents active in educational institutions?)

5. How were the questionnaires administered to the over 300 subjects? Did the authors administer them and provide details, or were they assisted by other people? Were those people able to provide additional information, as clearly as the authors?! Why do I ask this? Because in the “Acknowledgments” one of the authors thanks colleagues and students who helped? Or are these “students” the subjects involved in the study?!! In this case, they should use the same term, “adolescents” everywhere! I ask the authors to clarify!

6. Were the adolescents involved in the research provided with information regarding their role in what they have to complete or answer?

7. Were the assessment tools validated on the population, of which the subjects of this study are part?

8. I think that the numerical data in the manuscript should all be in the same format (with two decimal places). Please check everywhere in the manuscript

9. If the study is about children / adolescents, I recommend eliminating the words "male", "female"

10. In table 2, p=0.000.??? cannot be checked because the additional documents are not in English

11. I did not clearly understand what the purpose of the research was (the authors should clarify): to highlight certain relationships between the components of social support or to build a model and propose this model? I ask the authors to clarify in the conclusions and at the end of the discussions

12. I believe that the limitations of the study are few and I recommend the authors to analyze all aspects and specify them clearly (the study, as it is proposed, does not provide enough details to be replicated, due to limitations of context, age, level of education of the respondents, etc. etc.)

13. The conclusions are presented schematically!! I think it is more academic for authors to formulate clearly, concisely and scriptively what the ideas that resulted are. If readers want to read the title, abstract and conclusions, to see the basic idea, they have nothing to understand in the conclusions, in the form in which they are presented. The scheme with arrows can be included in the discussions or in the form of diagrams or figures, but the conclusions must be written

14. In References, there are extremely many references from the Asian area, which could provide subjectivity to the argued aspects!! It is not a bad thing, but it orients the ideas towards a certain area of the world, with certain influences of education or thinking, which does not offer the possibility of generalizing the conclusions. If they still maintain it, then the authors should conclude with reference to a certain typology of population. Maybe even in the title the reference to teenagers from China appears, and then the purpose and discussions that orient the idea are much clearer.

15. How did the authors evaluate whether each question is relevant, clear and sufficient?

16. How were the items translated or adapted for the subjects? Who checked the validity of the translation?

17. I recommend carefully checking the references, some are not relevant to this article, because they do not target the same age or context, and others are on subjects with conditions, and the authors did not specify anything about this aspect regarding the subjects in this study!!!

Reviewer #3: 

Introduction

The article begins by mentioning the importance of "non-cognitive skills", but its relationship to "social-emotional skills" is unclear. It is recommended to add a transition sentence.

The third paragraph mentions the definition and impact of physical exercise, but suddenly inserts "Scientific participation in physical exercise..." This seems abrupt. It is recommended to make a clear transition in the previous sentence.

The citation format is inconsistent, some have spaces before the brackets, and some do not.

There are issues with the standardization of English punctuation marks. Some punctuation marks have no space after them and need to be fully checked.

The overall research design is too simple, the method is outdated, and lacks innovation.

Research Methods

2.1 Participants

The article mentioned the distribution and return of questionnaires, but did not describe the sampling method (e.g., random sampling, convenience sampling), which affects the reproducibility of the study.

2.2.1 Measurement of physical activity

The calculation (Duration × Intensity × Frequency) does not indicate whether it has been validated or whether other studies support the rationality of the formula.

2.3 Data processing

The number of bootstrap samples (usually 5000) and the type of confidence interval (e.g., bias-corrected 95% CI) were not specified.

Results and analysis

3.1 Common method bias test

"Unrotated exploratory factor analysis extracted 11 factors with eigenvalues >1, the largest of which explained 34.719% of the variance (<40%)."

It was not stated whether principal component analysis (PCA) or exploratory factor analysis (EFA) was used, and the method was vague.

3.2 Descriptive statistics and correlation analysis

Only significance (p < 0.01) was reported, and the strength of the correlation coefficient (r) was not stated, which suggests that it is additional.

The font size of the table is not uniform, and it is recommended to coordinate the reference format.

Discussion

4.1 Physical Exercise and Social-Emotional Competencies

“Physical exercise plays a key role in improving adolescents’ social-emotional competences.

Cross-sectional studies cannot prove causality and cautionary statements should be used.

4.2 Independent mediating role of social support

Literature comparison is weak and only cites “previous findings” without specifying which studies support them.

“Social support equips adolescents with basic social skills…” Specific skills (e.g., conflict resolution, empathy) are not specified.

4.3 Independent mediating role of peer relationships

The mixed use of "peer acceptance" and "peer relationships"; coordination is suggested.

Classical theories (e.g., Sullivan's interpersonal relationship theory) can be cited to explain the role of peer relationships.

4.4 Chain mediation model analysis

"Social support consistently enhances peer relationships"

Unexplained mechanism, how social support reduces peer conflict.

4.5 Limitations

No mention of sample representativeness (only three schools in Guangdong) and homology bias.

"Considering other potential variables, such as cognitive reappraisal..."

No explanation of the relationship of these variables to the existing model.

**Do you want your identity to be public for this peer review?** For information about this choice, including consent withdrawal, please see our Privacy Policy

Reviewer #1: No

Reviewer #2: **Yes: ** Cristina Ioana Alexe

Reviewer #3: No

---

## [Author Response · Author response to Decision Letter 1]

4 Aug 2025

Dear Prof. Elena del Pilar Jiménez-Pérez; and all reviewers,

We sincerely thank you for your thoughtful and constructive feedback on our manuscript, "The Impact of Physical Exercise on Adolescents' Social-Emotional Competence: The Chain Mediating Role of Social Support and Peer Relationships" (Manuscript Number: PONE-D-25-20277). We greatly appreciate the time and attention you have dedicated to reviewing our work, and your insights have been instrumental in guiding our revisions. We have carefully considered each comment, implementing corresponding adjustments that we believe significantly enhance the rigor and clarity of our research. I will address the issues raised by the editor one by one in the cover letter. The following are detailed responses to the questions raised by all the reviewers:

Reviewer 1

1. This study aims to explore the relationship between physical exercise and adolescent socio-emotional competence and to analyze the mediating effects of social support and peer relationships in the chain. An interesting idea, but not a new one. This manuscript could bring an additional novelty, if it presented the details of the research application correctly and completely.

Response:

We sincerely appreciate your recognition of this research topic. Although it is not particularly novel, we consider it a vital direction for continued exploration in this field. Furthermore, regarding your suggestion that methodological details should be accurately and comprehensively presented, we fully endorse this perspective. In response to your feedback, we have thoroughly addressed this point in subsequent revisions, placing emphasis on refining and clearly presenting the applied research details with precision.

2. In the Introduction section, I recommend these studies, even though they are applied to athletes, most of whom are adolescents and provide support for the authors' argument:

https://doi.org/10.1177/00315125211005235,
https://doi.org/10.3390/children9050753

https://doi.org/10.7717/peerj.12803,
https://doi.org/10.1177/00315125221135669

https://doi.org/10.3390/ijerph19031696

Response:

①The article "Out-of-School Sports Participation Is Positively Associated with Physical Literacy, but What about Physical Education? A Cross-Sectional Gender-Stratified Analysis during the COVID-19 Pandemic among High-School Adolescents" proposes that "adolescents' engagement in physical exercise enhances their physical literacy, which encompasses physical, emotional, social, and cognitive attributes." This directly substantiates our assertion that "through physical exercise participation, adolescents improve physical literacy, thereby influencing self-awareness development" (Lines 50-51).

②The study "Measuring Perceived Social Support in Elite Athletes: Psychometric Properties of the Romanian Version of the Multidimensional Scale of Perceived Social Support" demonstrates that "social support promotes adolescents' sports participation." This empirically validates our proposed mechanism wherein "social support facilitates adolescents' engagement in physical exercise, ultimately forming a virtuous cycle" (Lines 278-279).

3. L151: "A total of 427 questionnaires were distributed ...." ??? Were 427 questionnaires or a single questionnaire applied to 427 subjects? I don't think it would be possible to apply 427 questionnaires!!! I think it's a mistake in the authors' expression. I recommend correct adaptation everywhere in this section and throughout the manuscript. Example here too ” 316 valid questionnaires”???

Response:

We appreciate your identification of this ambiguity. Due to imprecise wording in our original description, we have revised the section as follows: (Lines 155-158):

A total of 427 integrated questionnaires containing four scales were distributed, yielding 427 responses. After excluding 111 invalid responses (e.g., incomplete or duplicate entries), 316 valid datasets were retained, representing a 74% validity rate.

4. L154 ”The sample included 164 males and 152 females...” – I think this formulation should be adapted (example, maybe: teenagers and teenage girls), because at the age of 14 subjects cannot be classified as men or women

Response:

We sincerely appreciate your valuable feedback on the manuscript's phrasing. After consulting relevant literature, we fully concur with your perspective and have implemented the suggested revision:

Revised Statement (Line 158):"...including 164 boys (51.90%) and 152 girls (48.10%)."

5. L155: ” ± 1.405 years.” ?? I think expressing withonly 2 decimal places would be ok! That is ” ± 1.40 years.”

Response:

We gratefully acknowledge your insightful suggestion. The text has been revised as follows:

Revised Statement (Line 158): The participants' mean age was 14.73 ± 1.40 years.

6. I can't find the criteria for inclusion and exclusion of subjects in the study? I recommend highlighting them in the manuscript

Response:

We appreciate the reviewer's suggestions. In response to your recommendations, we have added detailed inclusion and exclusion criteria for the study participants in Section " 2.1 Participants". Revised Section " 2.1 Participants" (Lines 153-155):

The inclusion criteria for subjects were as follows: (1) Students aged 10–19; (2) Physically healthy with no motor impairments; (3) No cognitive impairments and able to accurately understand questionnaires and instructions. Individuals not meeting these criteria were excluded.

7. L163: ” Physical Activity Rating Scale developed by Hashimoto(1990)” – why such an old instrument? 35 years old? The authors should argue, because today's adolescents have a completely different vision of social aspects, compared to respondents 35 years ago, when was the instrument validated? Was it validated on a specific population? (another question addressed to the authors)

Response:

We appreciate your question. Here is our detailed response to the issues raised:

①Regarding whether today's adolescents have a different perception of society compared to adolescents 35 years ago: We would like to clarify that the questions in the scale are relatively objective in nature, such as the intensity, duration, and frequency of physical activity. Therefore, differences in social perception would not affect the objectivity of the responses.

②Regarding whether the scale has been validated for specific populations: We confirm that the scale has been extensively validated for the adolescent population in numerous studies. We have also added the following content to the manuscript to provide further support: "Previous studies have demonstrated the applicability of this scale for adolescents [60, 61]."

The revised content is located in Lines 179-180 of the manuscript. Please review it.

References

60. Guo W, Shao W, Guo Y, Zheng L. Effect of HAES intervention on obese adolescents' eating disordertendency and physical exercise behavior: The mediating effect ofweight self-stigma. Journal of Physical Education. 2024;31(05):62-7. doi: 10.16237/j.cnki.cn44-1404/g8.2024.05.015.

61. Guo Y, Chen Z, Wang H, Fan H. The Two-way Relationship Between Physical Activity and CognitiveReappraisal:A Study Based on Long-term Tracking and Diary Studies. Journal of Beijing Sport University. 2025;48(05):120-9. Doi : 10.19582/j.cnki.11-3785/g8.2025.05.010.

8. In the "2. Research Methods" section, data should be provided regarding the duration of completion, where were these assessment instruments applied, as a location? How long did it take to complete them? Were the subjects assisted by an adult? Did the subjects understand, accurately, at their age, all the items that the 4instruments assess? Did they have any questions and to whom did they address them? How were the 4 instruments applied: in a single form or in separate forms? Were the subjects aware that they were completing 4 assessments or did they know that they were completing a single instrument with 4 aspects? Is this section unclear and does not provide criteria for the reproducibility of the study?

Response:

We appreciate your identification of the issues. After carefully reviewing the "2. Research Methods" section, we have identified the deficiencies and have made the following point-by-point revisions:

①Regarding the data collection time: We have added the following information: Adolescents were surveyed between November 1 and November 30, 2024. (Lines 152-153)

②Regarding the location where the questionnaire was filled out: Paper questionnaires were administered by the research team during self-study sessions in classroom settings. (Lines 159-160)

③Regarding the estimated time to complete the questionnaire: an estimated completion time of 10 minutes. (Line 164)

④Regarding whether participants received assistance from adults and how their questions were addressed: We have added the following information: Before completing the questionnaires, researchers thoroughly explained the purpose, provided clear instructions, and addressed participants’ questions or concerns. Therefore, participants did receive adult assistance and had their questions addressed. (Lines 160-162)

⑤Regarding whether participants could accurately understand the questions posed by the four scales: We have added the following information:�1�The four questionnaires were selected from authoritative sources appropriate for the age group and have been validated for good reliability and validity among adolescents in this age range. We have also included references for each scale's application in adolescent populations in recent years in the"2.2 Variable Measurement" section. �2�The participants are normally educated adolescents aged 10-19 years, who possess the cognitive ability and logical thinking skills to accurately understand the questions posed in the questionnaire.(Lines 179, 185, 192, and 199)

⑥Regarding the format of the four scales: We have clarified the following information: Subjects were required to complete an integrated questionnaire comprising four scales: The Physical Activity Rating Scale, The Adolescent Social Support Scale, The Student Peer Relationships Scale, and The Social-Emotional Competence Questionnaire, totaling 66 items with an estimated completion time of 10 minutes. (Lines 162-164)

⑦ Regarding the provision of standards for research reproducibility: We have added the following information: This study has detailed the application of the research, including the inclusion and exclusion criteria for participants, detailed questionnaire information, experimental data, and the entire research process, which may provide a basis for verifying the experimental results.

We hope these revisions address your concerns.

9. The supplementary documents are not in English, which does not allow the reviewers to analyze these aspects (for example, I would like to find out details about Research protocols and Data.xlsx to verify the validity of the data, but I do not know the language used by the authors). I believe that translating the supplementary documents into English is essential, especially since some reviewers do not know the language used by the authors. I do not see the point of sending the main manuscript in English, the platform asks me if the English used by the authors is ok, but I, as areviewer, cannot analyze most of the documents!!!!!

Response:

We appreciate your feedback. We apologize for the oversight that led to some documents not being converted to English format, which may have caused inconvenience during the review process. We have now re-uploaded the research protocol and data in English on the system.

10. This study does not allow for data validation or reproducibility... due to the lack of certain details and supporting documents in a language that reviewers are not required to know in this journal.

Response:

We appreciate your feedback. We apologize for the oversight that led to some documents not being converted to English format, which may have caused inconvenience during the review process. We have now re-uploaded the data in English on the system.

Reviewer 2

1. The Introduction section should include more aspects of social support through sports activities.

Response:

We appreciate your question. In response, we have added relevant content regarding the acquisition of social support through physical exercise in the Introduction section. We have also included statements about the chain mediation model of physical exercise → social support → peer relationships → social and emotional competence. The revised content is as follows:

One study showed that physical exercise can help adolescents obtain social support from important figures such as friends and family, and promote the positive development of peer relationships [18]. Furthermore, the development of peer relationships was proven to be positively correlated with social-emotional competence, and this relationship is more pronounced among adolescent groups [19]. (Lines 54-57)

References

19. Weiss MR, Duncan SC. The relationship between physical competence and peer acceptance in tie context of children's sports participation. Journal of Sport and Exercise Psychology. 1992;14(2):177-91.

20. Han X, Li H, Niu L. How does physical education influence university students’ psychological health? An analysis from the dual perspectives of social support and exercise behavior. Frontiers in Psychology. 2025;16:1457165

2. I ask the authors to argue the relevance of the instruments used, compared to other instruments for evaluating the aspects targeted, which are newer, more recent (The first evaluation instrument is from 1990!!);

Response:

We appreciate your question. The scales we have selected have been widely cited in recent years and have undergone extensive validation among adolescent populations, demonstrating high reliability and validity. This indicates that the chosen scales possess a high level of authority. Additionally, in Section"2.2 Variable Measurement," we have included references for each scale cited within the past two years.

The supplementary content is located in Lines 179, 185, 192, and 199 of the manuscript. Please review it and let us know if you are satisfied.

3. I recommend that the authors add to the manuscript the criteria for including and excluding subjects in the research group

Response:

We appreciate the reviewer's suggestions. In response to your recommendations, we have added detailed inclusion and exclusion criteria for the study participants in Section " 2.1 Participants". Revised Section " 2.1 Participants" (Lines 153-155):

The inclusion criteria for subjects were as follows: (1) Students aged 10–19; (2) Physically healthy with no motor impairments; (3) No cognitive impairments and able to accurately understand questionnaires and instructions. Individuals not meeting these criteria were excluded.

4. I ask the authors to providemore details in the Research Methods section about how, when, where, how long the practical intervention lasted? (application of the questionnaire or the 4 evaluation instruments). To specify whether the adolescent subjects knew that they would be part of a research that targets several aspects!? The level of education of the adolescents is relevant and should be specified (were all respondents active in educational institutions?)

Response:

We appreciate your comments. In response to the issues you raised, we have carefully reviewed the “2. Research Methods” section and identified the deficiencies. We have made the following point-by-point revisions:

①Regarding the survey implementation method: We have added the following information: This study distributed one integrated questionnaire (containing four scales) in paper form to each of the 427 adolescent participants and excluded invalid questionnaires according to the screening criteria. (Lines 155-164)

②Regarding the survey time: We have added the following information: Adolescents were surveyed between November 1 and November 30, 2024. (Lines 152-153)

③Regarding the survey location: We have added the following information: The survey was conducted in three high schools in Guangdong Province, China, paper questionnaires were administered by the research team

---

## [Decision Letter · Decision Letter 1]

30 Sep 2025

The Impact of Physical Exercise on Adolescents' Social-Emotional Competence: The Chain Mediating Role of Social Support and Peer Relationships

PONE-D-25-20277R1

Dear Authors,

We’re pleased to inform you that your manuscript has been judged scientifically suitable for publication and will be formally accepted for publication once it meets all outstanding technical requirements.

Kind regards,

Ashraf Atta Mohamed Safein Salem

Academic Editor

PLOS ONE

Additional Editor Comments (optional):

Manuscript ID: PONE-D-25-20277R1

Title: The Impact of Physical Exercise on Adolescents' Social-Emotional Competence: The Chain Mediating Role of Social Support and Peer Relationships

Dear Author(s),

Thank you for submitting your revised manuscript to PLOS ONE. The reviewers have carefully evaluated your responses and the updated version of your article.

Both reviewers are satisfied that you have adequately addressed their comments. Reviewer 1 noted that the manuscript has been improved in structure and clarity, while Reviewer 2 indicated that your responses resolved their concerns and that no further comments were necessary.

In light of the reviewers’ recommendations and my own assessment, I am pleased to inform you that your manuscript has been accepted for publication in PLOS ONE.

We congratulate you on this accomplishment and look forward to the contribution your work will make to the field.

Sincerely,

Reviewers' comments:

Reviewer's Responses to Questions

**Comments to the Author**

Reviewer #1: All comments have been addressed

Reviewer #2: All comments have been addressed

2. Is the manuscript technically sound, and do the data support the conclusions?

Reviewer #1: Yes

Reviewer #2: Yes

3. Has the statistical analysis been performed appropriately and rigorously?

Reviewer #1: Yes

Reviewer #2: N/A

4. Have the authors made all data underlying the findings in their manuscript fully available?

Reviewer #1: Yes

Reviewer #2: Yes

5. Is the manuscript presented in an intelligible fashion and written in standard English?

Reviewer #1: Yes

Reviewer #2: Yes

Reviewer #1: The authors responded to my and the other reviewers' requests, clarified the issues in question, and adapted the manuscript into a better structured form. I have no further comments.

Reviewer #2: Dear authors,

In the review of an article, there are always other observations and questions, or recommendations. I, personally, will not make any more comments or observations, because you have answered my requests and I would risk, by other requests, diluting the direction and central idea of your research.

I wish you success and to develop the idea on a wider population, perhaps on a national level, so that you can generalize, at least at the level of children in your country. It is known that each region of a country also has different cultural contexts, education with differences, so that you cannot generalize a study on a region (you have now identified the limit, in your article).

**Do you want your identity to be public for this peer review?** For information about this choice, including consent withdrawal, please see our Privacy Policy

Reviewer #1: **Yes: ** Dan Iulian Alexe

Reviewer #2: **Yes: ** Cristina Ioana Alexe

---

## [Editor Report · Acceptance letter]

PONE-D-25-20277R1

PLOS ONE

Dear Dr. Zeng,

I'm pleased to inform you that your manuscript has been deemed suitable for publication in PLOS ONE. Congratulations! Your manuscript is now being handed over to our production team.

Kind regards,

on behalf of

Dr. Ashraf Atta Mohamed Safein Salem

Academic Editor

PLOS ONE